# Differentially Private Bayesian Persuasion

Submission Id: 394

## Abstract

The tension between persuasion and privacy preservation is common in real-world settings. Online platforms should protect the privacy of web users whose data they collect, even as they seek to disclose information about these data (e.g., to advertisers). Similarly, hospitals may share patient data to attract research investments with the obligation to preserve patients' privacy. To address these issues, we study Bayesian persuasion under differential privacy constraints, where the sender must design an optimal signaling scheme for persuasion while guaranteeing the privacy of each agent's private information in the database. To understand how privacy constraints affect information disclosure, we explore two perspectives within Bayesian persuasion: one views the mechanism as releasing a posterior about the private data, while the other views it as sending an action recommendation.

The posterior-based formulation leads to privacy-utility trade-offs, quantifying how the tightness of privacy constraints impacts the sender's optimal utility. For any instance in a common utility function family and a wide range of privacy levels, a significant constant gap in the sender's optimal utility can be found between any two of the three conditions: $\epsilon$-differential privacy constraint, relaxation $(\epsilon, \delta)$-differential privacy constraint, and no privacy constraint. We further geometrically characterize optimal signaling schemes under popular privacy constraints ($\epsilon$-differential privacy, $(\epsilon, \delta)$-differential privacy and Rényi differential privacy), which turns out to be equivalent to finding concave hulls in constrained posterior regions. Finally, we develop polynomial-time algorithms for computing optimal differentially private signaling schemes.

## CCS Concepts

• **Theory of computation → Algorithmic game theory**.

## Keywords

Bayesian persuasion, Differential privacy, Information design

**ACM Reference Format:**

Anonymous Author(s). 2024. Differentially Private Bayesian Persuasion. In .
ACM, New York, NY, USA, 16 pages.

## 1 Introduction

Online advertising platforms frequently hold auctions for advertising space and provide advertisers with user data to inform bidding decisions. However, as platforms prioritize their own profits over

advertisers', they must carefully design data publication mechanisms that incentivize advertisers to purchase ad space regardless of advertisers' profits.

This is a typical real-world situation of Bayesian persuasion, a model formalized by Kamenica and Gentzkow [31] where one party called *receiver* relies on information owned by another party called *sender* for decision-making. The informed sender can strategically design and commit an information-releasing mechanism, which is also called a *signaling scheme*, to persuade the receiver to act in the sender's interest.

Yet in releasing information to persuade advertisers, platforms may disclose users' private data, including age, gender, location, and interests. Thus, platforms must balance two competing goals: releasing helpful data to persuade purchase and preserving user privacy. Similar persuasion-privacy tensions arise in other examples. Hospitals may share patient data to attract research investments while preserving patients' privacy. Communities may publicize aggregated health statistics to encourage healthy habits without releasing details of specific residents.

In this paper, we provide a framework to study Bayesian persuasion under privacy constraints. To discuss data publication mechanism design, we model information owned by the sender, a.k.a. *state of nature*, as a database of binary trait responses from several agents. For privacy constraints, we adopt differential privacy, which is a standard privacy notion for information release. In general, it bounds changes in output distribution induced by small input alterations, such as changing a single user's data. Therefore, the signaling scheme committed by the sender should provide differential privacy guarantees for each agent's data in the state.

Under our differentially private Bayesian persuasion framework, we will illustrate what the optimal signaling scheme looks like under different types of differential privacy constraints, discuss how the tightness of privacy constraints influences the sender's optimal utility, and show how to efficiently compute the optimal scheme. The problem can be modeled as two equivalent forms of programming: one views it as the sender sending an action recommendation to the receiver and another as the sender releasing a posterior belief to transform the receiver's prior belief about an unknown state. By toggling between these two programs, we geometrically characterize and efficiently compute the optimal differentially private signaling scheme for persuasion.

### 1.1 Outline of Main Results

We primarily focus on the classic Bayesian persuasion setting proposed by Kamenica and Gentzkow [31] featuring one sender and one receiver, while we also provide some results for a more general multi-receiver setting. For privacy constraints, we mainly consider the constraints of pure differential privacy (or $\epsilon$-DP) and approximate differential privacy (or $(\epsilon, \delta)$-DP) [23, 24], but also extend results to Rényi differential privacy [41]. We model the state in the Bayesian persuasion model as the dataset of several agents, each

with a sensitive binary type. Neighboring datasets differ by only one agent's type, with all other agents' types remaining the same. The signaling scheme designed by the sender should guarantee the privacy of each agent by preserving differential privacy between all neighboring datasets.

We begin by considering single-agent cases where both the state and action spaces are binary. Using posterior-based programming, the optimal signaling scheme is straightforward to compute by restricting the region where the posterior satisfies the privacy constraints. However, feasible regions diverge significantly between pure and approximate differential privacy.

Our first main result analyzes the inherent tradeoff between privacy and utility, quantifying how increasingly stringent privacy guarantees impact the optimal utility achievable by the sender. For any instance in a class of utility functions with diverse application scenarios such as the advertising example introduced above, we show a significant constant utility gap for the sender between pure and approximate differential privacy across a wide range of privacy parameters (Theorem 3.4). Furthermore, a similar gap persists between differential privacy and no privacy constraint. Our results suggest that even slight relaxations from pure to approximate differential privacy significantly widen the feasible region and achievable payoffs. This sensitivity suggests the sender could pursue looser privacy constraints to enable substantially more utility.

We then consider more general cases with multiple agents where the state is an $n$-dimensional binary vector. Under $\epsilon$-DP, the sender faces a standard Bayesian persuasion problem. The posterior-based programming suggests that the sender should choose a distribution over posterior beliefs that the expectation matches the prior belief, to maximize the expectation of a special objective function parameterized by posteriors. Also, all posteriors in support of distribution should satisfy a set of linear privacy constraints. The characterization from Schmutte and Yoder [45] extends naturally, which is to find the concave hull of the objective function over the posterior space, restricted to a certain region, in order to determine the optimal signaling scheme.

However, when $\delta > 0$, the situation becomes much more complex. The main issue is that feasible regions now depend jointly across posteriors rather than admitting fixed constraints. Our second main result (Theorem 4.7) relates privacy constraints to ex ante constraints introduced in [5]. By including the privacy constraint dimensions and expanding the posterior space, we can characterize the optimal signaling scheme with the concave hull in certain regions. Also, Rényi differential privacy permits analogous characterization.

Our third main result turns to algorithmic aspects of Bayesian persuasion, exploring the more general multi-receiver setting. One potential concern with the geometric characterization is the exponential growth of the state space with the number of agents, leading to exponentially more variables in the programming and making it hard to find an efficient algorithm. However, real-world persuasion usually depends only on population statistics, not specific data points. Schmutte and Yoder [45] introduce a simplified oblivious signaling family where payoffs rely solely on statistic values rather than detailed data sources. Under natural assumptions, oblivious schemes optimally persuade compared with all possible schemes,

collapsing the state space polynomially. We show this reduction result also holds in our Bayesian persuasion model.

However, to establish our third main result (Theorem 5.2), another concern is that the signal space has to be exponential in the number of receivers, leading to intractably many variables in the action-based programming. We find an efficient separation oracle for its dual program under mild assumptions, and then oblivious signaling provides a tractable approach to multi-receiver differential privacy persuasion.

## 1.2 Related Work

First introduced by Kamenica and Gentzkow [31], the study of Bayesian persuasion has been driven by many real-world applications including criminal justice [31], security [42, 48], rooting [9], recommendation systems [39], auctions [10, 26], voting [1, 14] and queuing [36]. Our work is motivated by online advertising auctions, where the platform wants to persuade advertisers to buy advertising spaces. In these applications and others, more factors need to be considered, leading to many variants of the original model, such as limited communication capacity [28, 35], ordered state and action spaces [40], non-linear preference [34], the need of purchasing and selling information for sender [6, 7, 21], lack of knowledge about receivers [4, 12, 22] and so on (see surveys [8, 17, 30]).

The sender's problem of optimally releasing information about the state to maximize her payoff can be formulated in three main ways. The first models this as a choice of posterior distributions that are convex combinations of the prior [5, 16, 21, 31]. The second assumes the sender recommends an action as a function of the underlying state, which the receiver should find in their interest to follow [18, 19, 27, 33, 44, 46]. The third does not explicitly identify the optimal signal structure but rather characterizes the sender's optimal expected payoff via a concave envelope [37, 38]. Our work employs both the first formulation, to help characterize optimal signaling schemes, as well as the second, to enable efficient computation of optimal schemes. Following Dughmi and Xu [19], there are also many works considering algorithmic aspects of Bayesian persuasion, studying in the algorithmic game theory and theory of computation literature [2, 3, 12, 13, 15, 20, 28, 29, 47].

The most closely related work is Schmutte and Yoder [45], which considers differential privacy for information design problems. The key distinction from our setting is their focus on maximizing the receiver's utility. In other words, the sender's and receiver's utilities align, so the sender chooses a privacy-preserving mechanism to maximize value for end users. In contrast, our setting assumes misaligned sender and receiver utilities, which combined with the privacy constraints, leads to a more complex optimal mechanism. We also discuss different types of differential privacy and their influence on optimal signaling, while Schmutte and Yoder [45] only considers the standard $\epsilon$-DP.

Another relevant thread studies differential privacy, initiated by Dwork et al. [24]. Dwork et al. [23] proposed the widely-used relaxation $(\epsilon, \delta)$-DP, which provides comparable privacy protection as pure $\epsilon$-DP [32] but enables substantially more useful analyses. Dwork and Rothblum [25] put forth concentrated differential privacy, while Bun and Steinke [11] introduce zero-concentrated

differential privacy using Rényi divergence to capture privacy requirements [43]. Building on this, [41] proposes Rényi differential privacy, closely related to zero-concentrated differential privacy but focusing on single moments of the privacy variable.

## 2 Model and Preliminaries

### 2.1 Basic Setting

We consider a standard information disclosure setup as widely studied in the differential privacy literature. A database contains $n$ agents, each with a sensitive binary type $\theta_i \in \{0, 1\}$ (e.g., the agent is a targeted user of the advertiser or not). Let $\theta = (\theta_1, \ldots, \theta_n)$ denote the profile containing all agents' types. We also refer to $\theta$ as the *state* of the database. Let $\Theta \subseteq \{0, 1\}^n$ contain all possible states and $\mu \in \Delta(\Theta)$ denote a common prior belief over the states. Notably, $\mu$ may have a support that is exponential in $n$.

A *sender* looks to release information about the state $\theta$ by designing a *signaling scheme*. One or multiple *receivers* will use the released information for their own decision-making. Use $t$ to denote the number of receivers. Formally, before observing the state, the sender designs a signaling scheme $(S, \pi)$, sending separate signals $s_1, \cdots, s_t$ to each receiver, where $S$ is some countable set of *signals* and $\pi : \Theta \to \Delta(S^t)$ is a signaling scheme which maps each state to a distribution over signals in $S$, with $\pi(s \mid \theta)$ representing the probability of sending signal $s = (s_1, \cdots, s_t)$ when the state is $\theta$. Similar to $\mu$, the information releasing mechanism $\pi$ is also assumed to be publicly known.

After observing the signal $s^\star$, receiver $i$ updates its belief to the posterior $q_{s^\star}(\theta^\star) = \sum_{s:s_i=s^\star} \mu(\theta^\star)\pi(s \mid \theta^\star)/\sum_{\theta \in \Theta} \sum_{s:s_i=s^\star} \mu(\theta)\pi(s \mid \theta)$, which represents the probability that the true state is $\theta^\star$ conditioned on observing signal $s^\star$. The utility of receiver $i$ can be represented as a function $u_i : A \times \Theta \to \mathbb{R}$, depending on his action and the underlying state. Receiver $i$ then selects an action $a_i$ that maximizes its expected utility from the finite set $A$ based on this posterior belief. Mathematically, $a_i = \arg\max_{a \in A} \mathbb{E}_{\theta \sim q_{s^\star}}[u_i(a, \theta)]$. The sender has a different utility function $v : A^t \times \Theta \to \mathbb{R}$, depending on each receiver's action and the underlying state. This gives rise to the sender's strategic information releasing (also known as persuasion [30]) due to misaligned incentives. We use $u_1(a_1, \theta), \cdots, u_t(a_t, \theta)$, $v(a_1, a_2, \cdots, a_t, \theta)$ to denote the receivers' and sender's utilities, respectively.

For the main part, we also assume the sender's utility can be expressed as the sum of each receiver's utility individually. Mathematically, $v(a_1, a_2, \cdots, a_t, \theta) = \sum_i v_i(a_i, \theta)$. Therefore, the problem can be easily reduced to the single receiver setting by designing the signaling scheme independently for each receiver. For the simplicity of presentation, we consider the single receiver in the following and come back to general cases in Section 5.

### 2.2 Privacy Constraints to Information Disclosure

When releasing information to the receiver, the sender must guarantee the privacy of each agent hence cannot release too much information about each agent's type $\theta_i$. To quantify the privacy loss, we adopt standard privacy definitions from the literature of differential privacy.

The definitions of differential privacy are based on the notion of adjacent state pairs. Specifically, we say $\theta, \theta'$ adjacent if and only if they have only one different bit, that is $(\theta, \theta') \in X \Leftrightarrow \exists i, \theta_i \neq \theta'_i, \theta_{-i} = \theta'_{-i}$. $X$ is the set of all adjacent pairs here.

We also investigate scenarios where privacy protection is not required for every bit, a setting referred to as *partial privacy* in this paper[1]. We define a set $M \subseteq [n]$ to represent all bits in the state that require privacy protection and adjust the definition of adjacent pairs to $(\theta, \theta') \in X_M \Leftrightarrow \exists i \in M, \theta_i \neq \theta'_i, \theta_{-i} = \theta'_{-i}$. $X_M$ is the set of all adjacent pairs here. For simplicity of presentation, we still assume that the sender needs to protect privacy for all bits in the rest of the paper, i.e. $M = [n]$, and all of our results can be easily extended to the partial privacy setting.

Our main focus will be $(\epsilon, \delta)$-DP as well as an important special case of it simply by setting $\delta = 0$, i.e., the $\epsilon$-DP.

**Definition 2.1** $((\epsilon, \delta)$-DP [23]). *A signaling scheme $(S, \pi)$ preserves $(\epsilon, \delta)$-DP if for all subsets of signals $W \subset S$ and all $(\theta, \theta') \in X$, we have[2]*

$$\sum_{s \in W} \pi(s|\theta) \leq e^\epsilon \sum_{s \in W} \pi(s|\theta') + \delta.$$

*Specifically, when $\delta = 0$, it leads to a special case $\epsilon$-DP that*

$$\pi(s|\theta) \leq e^\epsilon \pi(s|\theta'), \text{ for any } s \in S \text{ and } (\theta, \theta') \in X.$$

For clarity, we always assume $\delta > 0$ when using $(\epsilon, \delta)$-DP and the special case of $\delta = 0$ is denoted as $\epsilon$-DP.

Besides the most common definition of $(\epsilon, \delta)$-DP, we also extend our results to Rényi differential privacy, which proves useful for theoretical analysis across a broad spectrum of problems. The definitions and results are delayed to Appendix A.

### 2.3 Sender's Objective

When choosing the signaling scheme under $(\epsilon, \delta)$-DP, the sender maximizes the expected value of his payoff. Formally, the sender solves Program 1[3].

Here the objective function is the expected utility of the sender under the signaling scheme $(S, \pi)$. The first constraint line shows that the expected utility of the receiver taking action $a_s$, upon observing signal $s$, must be greater than all other alternative actions. The third and fourth constraint guarantees the feasibility of the signaling scheme.

Note that Program (1) is almost the standard LP formulation for optimal Bayesian persuasion (see, e.g., [19]). The only new component is the second constraint, which guarantees $(\epsilon, \delta)$-DP. However, this difference is non-trivial. Indeed, the number of inequalities expressed in the second constraint is exponential in the number of signals since the DP constraint has to hold for every subset of signals $W$.

---

[1]The range of privacy guarantees varies in real-world examples. In advertising auctions, for instance, some users may consent to data sharing, obviating platform privacy obligations. Databases may also contain non-sensitive attributes like the size, location, and stability of ad spaces, and only user traits require privacy protection.
[2]For partial privacy, we only need to substitute $X_M$ for $X$. This also holds for other similar equations in the following.
[3]For the special case $\epsilon$-DP, privacy constraints can be written as: $\pi(s|\theta) \leq e^\epsilon \pi(s|\theta')$, for all $s \in S$ and $(\theta, \theta') \in X$.

$$\max_{(S,\pi)} \quad \sum_\theta \sum_s \pi(s|\theta)\mu(\theta)v(a_s,\theta)$$

$$\text{s.t.} \quad \sum_\theta u(a_s,\theta)\pi(s|\theta)\mu(\theta) \geq \sum_\theta u(a',\theta)\pi(s|\theta)\mu(\theta),$$
$$\text{for all } s \in S, a' \in A$$

$$\sum_{s \in W} \pi(s|\theta) \leq e^\epsilon \sum_{s \in W} \pi(s|\theta') + \delta, \quad \text{for all } W \subset S, (\theta,\theta') \in X$$

$$\sum_s \pi(s|\theta) = 1, \quad \text{for all } \theta \in \Theta$$

$$\pi(s|\theta) \geq 0, \quad \text{for all } \theta \in \Theta, s \in S \tag{1}$$

*Alternative formulation in the posterior space.* It turns out that Program (1) can be reformulated in the form of posteriors. Let the receiver's posterior after seeing signal s be $q_s$. The receiver will choose action $a^\star(q_s) = \arg\max_{a \in A} \mathbb{E}_{\theta \sim q_s}[u(a,\theta)]$. The sender's utility can be written as $V(q_s) = \mathbb{E}_{\theta \sim q_s}[v(a^\star(q_s),\theta)]$. The problem can then be seen as the sender choosing a distribution $\tau$ over posteriors that the expectation matches the prior, and maximizing the expectation of $V$ [31], while also adhering to specified privacy constraints. Mathematically, the reorganized program is[4]

$$\max_\tau \quad \mathbb{E}_{q_s \sim \tau}[V(q_s)]$$

$$s.t. \quad \mathbb{E}_{q_s \sim \tau}[q_s(\theta)] = \mu(\theta), \quad \text{for all } \theta \in \Theta$$
$$\sum_{q_s \in Q} \frac{q_s(\theta)\tau(q_s)}{\mu(\theta)} \leq e^\epsilon \sum_{q_s \in Q} \frac{q_s(\theta')\tau(q_s)}{\mu(\theta)} + \delta,. \tag{2}$$
$$\text{for all } Q \subset supp(\tau) \text{ and } (\theta,\theta') \in X$$

In the following, we will selectively use Program (1) and Program (2) as needed based on the results.

## 3 Privacy-Utility Tradeoffs

Under the imposed privacy constraints, the sender's expected utility will be restricted as the information can no longer be disclosed freely. In this section, we investigate how different privacy constraints, including $\epsilon$-DP, $(\epsilon,\delta)$-DP, or no privacy, impact the sender's optimal signaling scheme and expected utility.

We start with a simple case where there is only one agent with a binary type $\theta \in \{0,1\}$ and the action space is binary. The prior $\mu$ represents the probability that the agent's type is 1. For a large family of utility functions satisfying modest natural assumptions stated later, we show significant utility gaps exist between any two of three types of constraints (Theorem 3.4, Proposition 3.6, Proposition 3.7, Proposition 3.9). Such utility gaps are caused by essential difference in the geometry of the feasible regions under $\epsilon$-DP versus $(\epsilon,\delta)$-DP (Proposition 3.1).

All missing proofs in this section can be found in Appendix C.

### 3.1 Warm-up: Characterization for the Single Agent Case

We first show that using two signals is enough to give an optimal signaling scheme in the single-agent case. For the special case $\epsilon$-DP, this means the two posteriors cannot be too far away from the prior. However, under $(\epsilon,\delta)$-DP, the feasible region of posteriors

becomes more complex. Specifically, the feasible region for one posterior probability depends on the realization of the other posterior probability. To demonstrate this distinction, we depict the feasible regions under 0.2-DP and $(0.2,0.01)$-DP in Figure 1.

**Proposition 3.1.** *Consider any signaling scheme for the single-agent situation. Then we have*

- *There always exists an optimal signaling scheme that uses at most two signals.*
- *The scheme preserves $\epsilon$-DP if and only if the posterior $q_s$ of any induced signal s is in the interval $[\mu/(e^\epsilon - e^\epsilon\mu + \mu), \mu/(e^{-\epsilon} - e^{-\epsilon}\mu + \mu)]$.*
- *The scheme preserves $(\epsilon,\delta)$-DP if and only if the two posteriors $q_{s_1} \leq q_{s_2}$ satisfy the following inequalities:*

$$0 \leq q_{s_1} \leq \mu \leq q_{s_2} \leq 1, \quad q_{s_1}(C_1 q_{s_2} - C_2) \geq C_3 q_{s_2} - \mu^2,$$
$$q_{s_1}(C_4 q_{s_2} - C_5) \geq C_6 q_{s_2} - e^\epsilon \mu^2,$$

*where $C_1 = e^\epsilon - \mu e^\epsilon + \mu$, $C_2 = \mu(e^\epsilon - \mu e^\epsilon + \mu) + \delta\mu(1-\mu)$, $C_3 = \mu - \delta\mu(1-\mu)$, $C_4 = 1 - \mu + e^\epsilon\mu$, $C_5 = e^\epsilon\mu + \delta\mu(1-\mu)$, $C_6 = \mu(1 - \mu + e^\epsilon\mu) - \delta\mu(1-\mu)$.*

**Figure 1: Feasible regions of posteriors when $\mu = 0.5$. The pink region represents possible posteriors $(q_{s_1}, q_{s_2})$ under $0.2$-DP. The total area of pink and blue regions represents possible posteriors under $(0.2, 0.01)$-DP.**

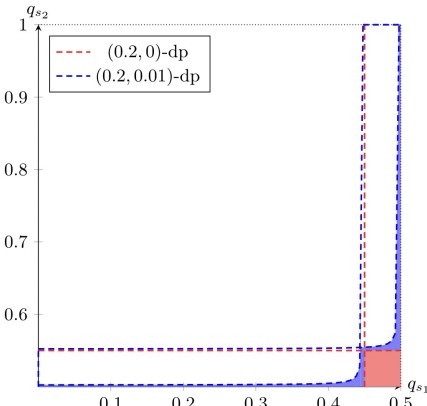

Proposition 3.1 shows the feasible posterior region $C$ under different privacy constraints. We then present a proposition to show how to compute the maximum utility and the optimal signaling scheme for the sender given the feasible region. For Program (2), all $(q_{s_1}, q_{s_2}) \in C$ satisfy privacy constraints. We then only need to maximize $\mathbb{E}_{q_s \sim \tau}[V(q_s)]$ with the constraint $\mathbb{E}_{q_s \sim \tau}[q_s] = \mu$.

**Proposition 3.2.** *Given the feasible posterior region $C$ for $(q_{s_1}, q_{s_2})$, the maximum utility for sender is*

$$\max_{(q_{s_1}, q_{s_2}) \in C} \frac{(q_{s_2} - \mu)V(q_{s_1}) + (\mu - q_{s_1})V(q_{s_2})}{q_{s_2} - q_{s_1}}$$

*and the optimal signaling scheme is inducing $q_{s_1}, q_{s_2}$ which makes the above equation achieve maximum value.*

---

[4]Also, for the special case $\epsilon$-DP, privacy constraints can be written as $\frac{q_s(\theta)\mu(\theta')}{q_s(\theta')\mu(\theta)} \leq e^\epsilon$, for all $q_s \in supp(\tau)$ and $(\theta,\theta') \in X$.

## 3.2 Utility Gaps

From Proposition 3.1, we observe that though $\epsilon$-DP can be seen as a special version of $(\epsilon, \delta)$-DP when $\delta = 0$, their feasible posterior regions can be fundamentally different. When $\delta > 0$, it is possible to have one posterior at an extreme value like 0 or 1, although this would make the other posterior very close to the prior, thus limiting the weight of the extreme posterior. Therefore, a key question is whether this difference in the shape of feasible areas can lead to a large difference in the sender's maximum utility. We consider a large family of utility functions, exemplifying the nature of the utility functions in the advertising auction example, while also well applied to many other real-world scenarios. A significant gap in the sender's maximum utility under $\epsilon$-DP and $(\epsilon, \delta)$-DP could be found in this family of utility functions.

We first define the utility function family. Here we use $v$ to denote the sender's utility function and $u$ to denote the receiver's function.

**Definition 3.3.** *Utility functions* $(v, u) \in \Gamma$ *if*

- *$v$ is state-independent, which means it is only decided by the receiver's action.*
- *$v$ equals 1 facing action 1 and 0 facing action 0.*
- *The receiver chooses action 0 when the posterior is smaller than some value, otherwise, she chooses action 1. Let the threshold be $t$.*

Note that the first requirement actually means transparent motives [38], which is natural in our advertising example and many other real-world scenarios. The second one is trivial since we can easily reduce other utilities to this easy case by scaling. The third one is set to avoid that the receiver will always take one action regardless of the posterior.

For simplicity of presentation, we define $\mathrm{OPT}_{v,u,\mu}(\epsilon, \delta)$ as the sender's optimal utility under $(\epsilon, \delta)$-DP when the sender and receiver's utility functions are $(v, u)$ and the prior is $\mu$. Then $\mathrm{OPT}_{v,u,\mu}(\epsilon, 0)$ denotes optimal utility under $\epsilon$-DP. Also, $\mathrm{OPT}_{v,u,\mu}(\infty, \infty)$ indicates the case without privacy constraints.

*Utility gap between $\epsilon$-DP and $(\epsilon, \delta)$-DP.* An additive utility gap can be found for any utility functions in $\Gamma$ between $\epsilon$-DP and $(\epsilon, \delta)$-DP. Also, it holds between $\epsilon_1$-DP and $(\epsilon_2, \delta)$-DP when $\epsilon_1$ is slightly larger than $\epsilon_2$.

**Theorem 3.4.** *For any* $(v, u) \in \Gamma$, *if* $\epsilon_1 - \epsilon_2 \leq C\delta$ *for some* $C < 1$, *then there exists a* $\mu$ *such that* $\mathrm{OPT}_{v,u,\mu}(\epsilon_2, \delta) - \mathrm{OPT}_{v,u,\mu}(\epsilon_1, 0) \geq (e^{\epsilon_2} - 1)/((1 + \delta)e^{\epsilon_1 + \epsilon_2} - 1)$.

To prove this theorem, we first strategically choose $\mu$ such that the sender is unable to obtain positive utility under $\epsilon_1$-DP. However, under $(\epsilon_2, \delta)$-DP, the sender can set one posterior equal to $t$ while ensuring the other posterior is sufficiently far from the prior, resulting in the maximum attainable weight of positive utility.

In practice, $\delta$ typically needs to be sufficiently small since the sender could otherwise directly release one bit of information with probability $\delta$ [32]. Therefore, when $\delta$ is small but $\epsilon_1$ and $\epsilon_2$ are moderately large, as is common, the above theorem demonstrates a constant utility gap.

**Corollary 3.5.** *For any* $(v, u) \in \Gamma$, *if* $\delta \leq 0.01, 0.01 \leq \epsilon_2 \leq \epsilon_1 \leq 1$ *and* $\epsilon_1 - \epsilon_2 \leq C\delta$ *for some* $C < 1$, *then there exists a* $\mu$ *such that* $\mathrm{OPT}_{v,u,\mu}(\epsilon_2, \delta) - \mathrm{OPT}_{v,u,\mu}(\epsilon_1, 0) \geq 0.25$.

Since in the proof of Theorem 3.4, we actually select a $\mu$ that makes the sender unable to get positive utility under $\epsilon$-DP. This actually implies an arbitrarily large gap between any instances of the two types of privacy constraints.

**Proposition 3.6.** *For any* $(u, v) \in \Gamma$, $\epsilon_1, \epsilon_2, \delta > 0$, *there exists a* $\mu$ *satisfying that* $\mathrm{OPT}_{v,u,\mu}(\epsilon_2, \delta)/\mathrm{OPT}_{v,u,\mu}(\epsilon_1, 0)$ *can be arbitrarily large.*

*Utility gap between $\epsilon$-DP and no privacy.* Also, when there is no privacy constraint, the sender can achieve better utility. There is also a large additive gap between $\epsilon$-DP and no privacy.

**Proposition 3.7.** *For any* $(v, u) \in \Gamma$, *there exists a* $\mu$ *such that* $\mathrm{OPT}_{v,u,\mu}(\infty, \infty) - \mathrm{OPT}_{v,u,\mu}(\epsilon, 0) \geq 1/((1 - t)e^{\epsilon} + t)$.

The above proposition implies a constant gap for all $\epsilon < 1$.

**Corollary 3.8.** *For any* $(v, u) \in \Gamma$, *if* $\epsilon \leq 1$, *there exists a* $\mu$ *such that* $\mathrm{OPT}_{v,u,\mu}(\infty, \infty) - \mathrm{OPT}_{v,u,\mu}(\epsilon, 0) \geq 0.37$.

*Utility gap between $(\epsilon, \delta)$-DP and no privacy.* Moreover, an additive gap occurs between $(\epsilon, \delta)$-DP and no privacy.

**Proposition 3.9.** *For any* $(v, u) \in \Gamma$, *there exists a* $\mu$ *such that*

$$\mathrm{OPT}_{v,u,\mu}(\infty, \infty) - \mathrm{OPT}_{v,u,\mu}(\epsilon, \delta) \geq \frac{1}{2} - \min\left\{ \frac{(1 - \mu)(\delta + e^{\epsilon} - 1)}{(1 - \mu)(2e^{\epsilon} - 1) + \mu}, \right.$$
$$\left. \frac{(1 - \mu)\delta}{\max\{2(1 - \mu)\delta, (1 - \mu)(2 - e^{\epsilon}) + e^{\epsilon}\mu\}} \right\}.$$

The above proposition implies a constant gap when $\delta$ is small and $\epsilon$ is not too big.

**Corollary 3.10.** *For any* $(v, u) \in \Gamma$, *if* $\delta \leq 0.01, \epsilon \leq 0.5$, *there exists a* $\mu$ *such that* $\mathrm{OPT}_{v,u,\mu}(\infty, \infty) - \mathrm{OPT}_{v,u,\mu}(\epsilon, \delta) \geq 0.47$.

To better illustrate how the optimal signaling scheme and the sender's maximum utility differ under varying privacy constraints, we show an example here.

**Example 3.11.** *An auctioneer aims to sell advertising space that will be displayed to a single web user, to a particular advertiser. The auctioneer possesses binary data about whether this user is in the advertiser's target customer segment. The auctioneer's sole objective is that the advertiser purchases the ad space, that is $v(1, \theta) = 1, v(0, \theta) = 0$, for $\theta \in \{0, 1\}$.*

*For the advertiser, if he chooses not to buy the space, his utility is a constant 0. If he chooses to buy the space, his utility is 1 when the web user is the targeted customer and $-1$ when the web user is not.*

$$u(0, \theta) = 0, \text{ for } \theta \in \{0, 1\}, u(1, 0) = -1, u(1, 1) = 1.$$

*Then the sender's utility can be written as*

$$V(q) = \begin{cases} 0, & q < 0.5, \\ 1, & q \geq 0.5. \end{cases}$$

*There are several privacy considerations for the sender. One choice is that the signaling scheme needs to satisfy $\epsilon$-DP. Another choice is that it needs to satisfy $(\epsilon - \delta/2, \delta)$-DP. We also consider the condition of no privacy constraint. Here we only consider $\epsilon$ and $\delta$ less than 1.*

*The common prior $\mu$ that the user is a targeted customer is $(1 - \zeta)/(1 + e^{\epsilon})$. Here $\zeta$ is an arbitrarily small positive number.*

*Without privacy constraint, the optimal signaling scheme using two signals $s_1, s_2$ which satisfies $q_{s_1} = 0$, $q_{s_2} = 0.5$ can give a utility of $2(1 - \zeta)/(1 + e^{\epsilon})$.*

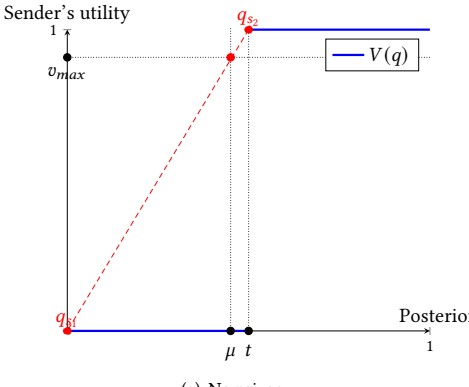

(a) No privacy

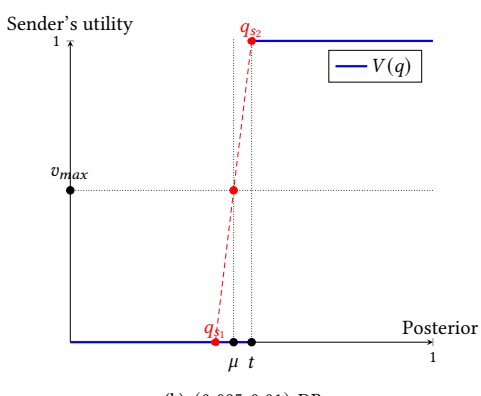

(b) $(0.095, 0.01)$-DP

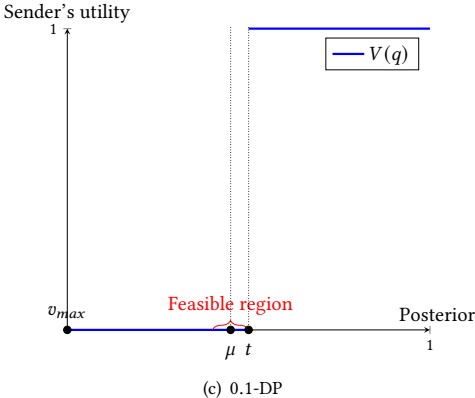

(c) 0.1-DP

**Figure 2: Different privacy constraints in Example 3.11.** $t = 0.5$ and $\mu = 0.475$. The optimal signaling schemes without privacy constraints and under $(0.095, 0.01)$-DP are shown in (a) and (b). Under 0.1-DP, no signaling scheme can guarantee a positive utility, and the feasible posterior region is shown in (c).

Under $(\epsilon - \delta/2, \delta)$-DP, the optimal signaling scheme using two signals $s_1, s_2$ which satisfies

$$q_{s_1} = \frac{\frac{e^\epsilon - 1 - 2\zeta}{e^\epsilon + \zeta} - \delta}{(\frac{1}{e^\epsilon + \zeta} + \frac{e^\epsilon}{1 - \zeta})(e^\epsilon - 1 - 2\zeta) - 2\delta}, \quad q_{s_2} = 0.5$$

can give a utility of

$$\frac{1 - (1 + \epsilon)q_{s_1}}{(0.5 - q_{s_1})(1 + e^\epsilon)}.$$

While under $\epsilon$-DP, the feasible area for the posterior is

$$\left[\frac{1 - \zeta}{e^{2\epsilon} + 1 + (e^\epsilon - 1)\zeta}, \frac{e^\epsilon(1 - \zeta)}{2e^\epsilon(1 - \zeta) + (e^\epsilon + 1)\zeta} < 0.5\right].$$

Therefore, no signaling scheme can be used to persuade the advertiser to buy, and the maximum utility of the advertiser is 0.

For a large range of $\epsilon$ and small $\delta$, a significant utility gap exists between any two of the above constraints. To give a numerical example, we set $\epsilon = 0.1$ and $\delta = 0.01$, then $\mu = 0.475(1 - \zeta)$. Here we omit $\zeta$ and use $\mu$ as 0.475.

Without privacy constraints, the sender can design the optimal signaling scheme $q_{s_1} = 0$, $q_{s_2} = 0.5$ to get a utility of 0.95, which is nearly the maximum utility. This is shown in Figure 2(a).

Adding $(0.095, 0.01)$-DP, the optimal signaling scheme $q_{s_1} = 0.445$, $q_{s_2} = 0.5$ only guarantees a utility of 0.541, which is about half of the utility under the condition without privacy constraints. This is shown in Figure 2(b).

However, under 0.1-DP, even if 0.1 is slightly larger than 0.095, there doesn't exist a signaling scheme to help the sender get positive utility. This is shown in Figure 2(c).

## 4 Characterization of Privacy-Constrained Bayesian Persuasion

Now we turn to general cases where there are multiple agents and $\theta$ is an n-dimensional binary vector. By using techniques in previous information design literature ([5, 16, 45]), we obtain general geometric characterization for different kinds of differential privacy constraints. For $\epsilon$-DP, the optimal signaling scheme is finding the concave hull of the objective function $V$ in a feasible region (Proposition 4.1). However, for $(\epsilon, \delta)$-DP privacy, there doesn't exist a fixed feasible region to find a concave hull. Through enlarging the posterior space and modifying objective function $V$, we can still relate the problem to a concave hull in a constrained region (Theorem 4.7). These geometrical characterizations also provide useful methods for determining whether the sender can benefit from persuasion (Corollary 4.2, Corollary 4.8).

To characterize the optimal signaling scheme and the maximum utility of the sender, we now give a formal definition of concave hull. We use $k$ to denote the number of states, and $m$ to denote the number of pairs of adjacent states in $X$ which is also the number of privacy constraints[5]. Note that in the most general setting, both $k$ and $m$ can scale exponentially. However, under reasonable assumptions stated later, we will show that both quantities can be restricted to polynomial growth (see Lemma D.2). The concave hull at $x$ of the function $V$ in a constrained region $K$ is defined as

$$\overline{V}(x \mid K) := \max_{Q \subseteq K} \left( \sum_{q \in Q} \tau(q)V(q) : \sum_{q \in Q} \tau(q)q = x, \sum_{q \in Q} \tau(q) = 1, \right.$$

$$\left. \tau(q) \geq 0 \text{ for all } q \in Q \right).$$

---

[5]For partial privacy, we replace $X$ with $X_M$.

We also use a function $S_V$ to represent the set $Q$ that make up the concave hull of $V$ at $x$:

$$S_V(x \mid K) \coloneqq \underset{Q \subseteq K}{\arg\max} \left( \sum_{q \in Q} \tau(q) V(q) : \sum_{q \in Q} \tau(q) q = x, \sum_{q \in Q} \tau(q) = 1, \right.$$
$$\left. \tau(q) \geq 0 \text{ for all } q \in Q \right).$$

*Characterization for $\epsilon$-DP.* Schmutte and Yoder [45] discuss the characterization of $\epsilon$-DP information design problem, where the sender's utility function is the same as the receiver's. We can directly extend Theorem 1 in [45] to obtain that the optimal signaling scheme for Bayesian persuasion under $\epsilon$-DP is related to the concave hull of $V$ in a region of $\epsilon$-DP posteriors, which is a closed convex polyhedron. Mathematically, the feasible region $K(\mu, \epsilon)$ is $\{q \in \Delta(\Theta) : q(\theta)\mu(\theta')/(q(\theta')\mu(\theta)) \leq e^\epsilon \text{ for all } (\theta, \theta') \in X\}$. Also, $k$ signals suffice for an optimal signaling scheme under $\epsilon$-DP.

**Proposition 4.1** (Schmutte and Yoder [45])**.** *For $\epsilon$-DP, there exists a valid optimal signaling scheme such that*

- *The scheme uses $|S_V(\mu \mid K(\mu, \epsilon))|$ signals, which is not larger than $k$.*
- *The scheme induces posteriors in set $S_V(\mu \mid K(\mu, \epsilon))$.*
- *The optimal utility of the sender is $\overline{V}(\mu \mid K(\mu, \epsilon))$.*

**Corollary 4.2.** *Sender benefits from persuasion under $\epsilon$-DP if and only if $\overline{V}(\mu \mid K(\mu, \epsilon)) > V(\mu)$.*

*Characterization for $(\epsilon, \delta)$-DP.* However, for $(\epsilon, \delta)$-DP, there doesn't exist such $K(\mu, \epsilon)$ as $\epsilon$-DP since the feasible region of one posterior depends on the specific value of other posteriors, which can be seen in Proposition 3.1. We first review some important results from previous work about constrained information design, which is a more general model. Here is the formulation [5].

$$\max_{\tau} \quad \mathbb{E}_{q \sim \tau}[f(q)]$$
$$s.t. \quad \mathbb{E}_{q \sim \tau}[q(\theta)] = \mu(\theta), \text{ for all } \theta \in \Theta$$
$$\mathbb{E}_{q \sim \tau}[g_i(q)] \leq c_i, \quad \text{for } i = 1, 2, \cdots, I$$
$$h_j(q) \leq d_j, \qquad \text{for all } q \in supp(\tau), j = 1, 2, \cdots, J$$
$$\tag{3}$$

where the objective is still to pick a distribution over posteriors that has expectation $\mu$ and satisfies $I + J$ extra constraints.

Here Babichenko et al. [5] actually define two general families of constraints: *ex ante* and *ex post*. A constraint of the latter type restricts the admissible values of a certain function of posteriors for every possible posterior, while a constraint of the former type restricts only the expectation of such a function. Here we use the posterior distribution $\tau$ to denote the signaling scheme.

**Definition 4.3** (Ex ante Constraints [5])**.** *An* ex ante constraint *on a signaling scheme $\tau$ is a constraint of the form $\mathbb{E}_{q \sim \tau}[g(q)] \leq c$ for continuous $g : \Delta(\Theta) \to \mathbb{R}$ and a constant $c \in \mathbb{R}$.*

**Definition 4.4** (Ex post Constraints [5])**.** *An* ex post constraint *on a signaling scheme $\tau$ is a constraint of the form $\forall q \in supp(\tau), h(q) \leq d$ for continuous $h : \Delta(\Theta) \to \mathbb{R}$ and a constant $d \in \mathbb{R}$.*

For upper semi-continuous objective function $f$, Doval and Skreta [16] prove that for every set of k states and m ex ante constraints, there exists a valid optimal signaling scheme with a support size of at most $k + m$.

**Lemma 4.5** ([16])**.** *Fix $k$ states, $m$ ex ante constraints, and no ex post constraints. Then either there exists an optimal valid signaling scheme with support size at most $k + m$ or the set of valid signaling schemes is empty.*

Note that $\epsilon$-DP constraints satisfy the requirement of ex post constraints. Babichenko et al. [5] establish a stronger bound that for $k$ states, no ex nate constraints and a set of ex post constraints, $k$ signals suffice for the optimal signaling scheme, which matches with Proposition 4.1.

For Program (3) without ex post constraints(i.e., $j = 0$), Babichenko et al. [5] connects the value of the program to the concave hull of a modified version of the objective function $f$, which is introduced next.

*A modified objective function.* Let $C = \left\{ (\tilde{\mu}, \tilde{c}) \in \Delta(\Theta) \times \mathbb{R}^I : g_I(\tilde{\mu}) \leq \tilde{c}_I \right\}$. That is, $C$ is the subset of $\Delta(\Theta) \times \mathbb{R}^I$ which satisfies a pointwise version of constraints in Program (3). Given $C$, define the function $f^g : \Delta(\Theta) \times \mathbb{R}^I \mapsto \mathbb{R} \cup \{\pm\infty\}$ as follows,

$$f^g(\tilde{\mu}, \tilde{c}) = f(\tilde{\mu}) - \delta(\tilde{\mu}, \tilde{c} \mid C),$$

where $\delta(\tilde{\mu}, \tilde{c} \mid C)$ is the indicator function of $C$, taking value 0 if $(\tilde{\mu}, \tilde{c}) \in C$ and $+\infty$ otherwise.

**Lemma 4.6** ([5])**.** *The value of Program (3) when $j = 0$ coincides with the value of concave hull of $f^g$ at $(\mu, c)$.*

We now turn to $(\epsilon, \delta)$-DP. Under $(\epsilon, \delta)$-DP, the program can be written as

$$\max_{\tau} \quad \mathbb{E}_{q_s \sim \tau}[V(q_s)]$$
$$s.t. \quad \mathbb{E}_{q_s \sim \tau}[q_s(\theta)] = \mu(\theta), \quad \text{for all } \theta \in \Theta,$$
$$\sum_{q_s \in Q} \frac{q_s(\theta)\tau(q_s)}{\mu(\theta)} \leq e^\epsilon \sum_{q_s \in Q} \frac{q_s(\theta')\tau(q_s)}{\mu(\theta)} + \delta,.$$
$$\text{for all } Q \subset supp(\tau) \text{ and } (\theta, \theta') \in X$$

Here the form of privacy constraints poses difficulty to the characterization since it has to traverse all possible subsets of posteriors. An equivalent but much easier form is

$$\mathbb{E}_{q_s \sim \tau}\left[ \max\left\{ 0, \left( \frac{q_s(\theta)}{\mu(\theta)} - e^\epsilon \frac{q_s(\theta')}{\mu(\theta')} \right) \right\} \right] \leq \delta, \text{ for all } (\theta, \theta') \in X.$$

A nice observation here is that $(\epsilon, \delta)$-DP exactly satisfies the form of ex ante constraints.

Note that $V$ is upper semi-continuous in our discrete setting. Also, releasing nothing but the prior implies the existence of a valid signaling scheme, then Lemma 4.5 shows an optimal signaling scheme under $(\epsilon, \delta)$-DP needs not to use more than $k + m$ signals. Also, Lemma 4.6 implies a geometrical characterization for $(\epsilon, \delta)$-DP, which connects the optimal signaling scheme to the concave hull of a modified version of the objective function $V$.

*A modified objective function for $(\epsilon, \delta)$-DP.* We modify the function $V$ to take two parts as input rather than a single posterior. The first part is the original posterior. The second part is an $m$-dimensional input corresponding to each adjacent state pair in the set $X$. Mathematically, the modified function $F : \Delta(\Theta) \times \mathbb{R}^m \mapsto \mathbb{R}$

is $F(q, \gamma) = V(q)$. As for $\epsilon$-DP, we also define a region for $(\epsilon, \delta)$-DP,

$$C(\mu, \epsilon) := \left\{ (q, \gamma) \in \Delta(\Theta) \times \mathbb{R}^m : \max \left\{ 0, \left( \frac{q(\theta)}{\mu(\theta)} - e^\epsilon \frac{q(\theta')}{\mu(\theta')} \right) \right\} \leq \gamma_{\theta, \theta'}, \right.$$
$$\text{any } (\theta, \theta') \in X \Big\}.$$

An example is shown in Figure 3.

Then we can characterize the optimal signaling scheme and maximum utility with the concave hull of $F$.

**Figure 3: The feasible region $C(\mu = 0.5, \epsilon = 0.2)$ for $(0.2, \delta)$-DP under different posteriors while $\theta \in \{0, 1\}$. Note that the region is $\delta$-independent. $\gamma_1$ corresponds to the adjacent pair $(\theta = 1, \theta' = 0)$ and $\gamma_2$ corresponds to the adjacent pair $(\theta = 0, \theta' = 1)$. $q$ represents the probability of state 1.**

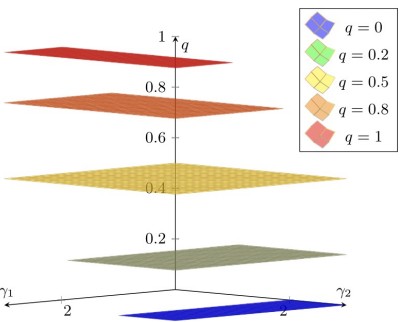

**Theorem 4.7.** *For $(\epsilon, \delta)$-DP, there exists a valid optimal signaling scheme such that*

- *The scheme uses $|S_F(\mu, \boldsymbol{\delta} \mid C(\mu, \epsilon))|$ signals, which is not larger than $k + m$. Here $\boldsymbol{\delta}$ is the $m$-dimension vector with all elements $\delta$.*
- *The scheme induces posteriors in set $S_F(\mu, \boldsymbol{\delta} \mid C(\mu, \epsilon))$.*
- *The optimal utility of the sender is $\overline{F}(\mu, \boldsymbol{\delta} \mid C(\mu, \epsilon))$.*

**Corollary 4.8.** *Sender benefits from persuasion under $(\epsilon, \delta)$-DP if and only if $\overline{F}(\mu, \boldsymbol{\delta} \mid C(\mu, \epsilon)) > V(\mu)$.*

Note that Corollary 4.2 and Corollary 4.8 provide a natural analysis of the condition where the sender can benefit from the persuasion. We can also present some alternative, indirect analyses, building on the approach of Kamenica and Gentzkow [31], see Appendix B.

## 5 Algorithmics of Privacy-constrained Bayesian Persuasion

In this section, we turn to algorithmic aspects of the persuasion problem, returning to the formulation in Program (1) with signal distributions as variables. We consider a general setting with multiple receivers, which means not assuming separate sender's payoffs from each receiver any more. Here one concern is the exponential growth of the state space, thus exponentially more variables and constraints with more agents. Another key concern is an exponential action space (namely signal space) in the number of receivers, inducing exponentially many variables. We show a polynomial

time algorithm under homogeneity assumptions (Theorem 5.2). All missing proofs can be found in Appendix D.

For the simplicity of presentation, we consider binary actions here, and the technique can be extended to constant many actions straightforwardly. Also, we use $t$ to denote the number of receivers.

We now present our main result, computing the optimal signaling scheme in polynomial time based on several reasonable assumptions.

When considering web users' data in advertising cases, users are often seen as homogeneous, both in terms of prior and utility functions. This means the sender and receiver only care about the aggregate statistics of the database rather than individual users' data. Let $\phi : \Theta \to \Omega$ be the projection that only considers the number of 1s in a given state[6]. That is, $\phi(\theta) := \sum_i \theta_i$. We further assume that the sender only cares about the number of 1 in the action profile. We denote by $T \subset \{1, \ldots, t\}$ the subset of receivers adopting action 1.

**Assumption 5.1.** *The prior $\mu$, sender's utility function $v$ and receivers' utility functions $u_1, \ldots, u_t$ satisfy*

- *If $\phi(\theta) = \phi(\theta')$, then $\mu(\theta) = \mu(\theta')$.*
- *If $\phi(\theta) = \phi(\theta')$, then $u_i(T, \theta) = u_i(T, \theta')$ and $v(T, \theta) = v(T, \theta')$ for any $i$ and $T$.*
- *If $|T_1| = |T_2|$, then $v(T_1, \theta) = v(T_2, \theta)$ for any $\theta$.*

**Theorem 5.2.** *Under Assumption 5.1, there is a polynomial time algorithm to compute the optimal signaling scheme.*

*Proof Sketch of Theorem 5.2.* We prove the theorem in three steps.

Step 1. We first show that using the same number of signals as the number of actions is enough to give an optimal signaling scheme. Then $2^t$ signals suffice, corresponding to selecting a subset of receivers to suggest action 1, with others suggested action 0.

Step 2. It is natural to consider the oblivious method [45], where the signal distributions are identical for any states $\theta$ and $\theta'$ such that $\phi(\theta) = \phi(\theta')$. That is, the signal distribution depends only on the projection $\phi(\theta)$. We then prove that there exists an oblivious privacy-preserving signaling scheme that is optimal among all privacy-preserving signaling schemes.

Step 3. For the program under the oblivious method, we consider its dual program and find an efficient separation oracle. Given the oracle, we can use the Ellipsoid method to obtain a vertex optimal solution to both program and its dual program in polynomial time.

## 6 Conclusion

In this work, we study the Bayesian persuasion model under differential privacy constraints. Our research sheds light on the critical questions of how to characterize and efficiently compute optimal signaling schemes under various notions of differential privacy. We also examine the inherent privacy-utility tradeoffs and extend our findings to more general settings involving partial privacy guarantees and multiple receivers. Promising directions for future work include exploring additional varieties of privacy constraints and incorporating increased complexities into the persuasion model, such as adding mediators.

---

[6]For partial privacy, the projection needs to consider the number of 1s among the sensitive bits under privacy protection and non-sensitive bits separately. Mathematically, $\phi_M(\theta) := (\sum_{i \in M} \theta_i, \sum_{i \notin M} \theta_i)$.

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

# A  Generalization to Rényi Differential Privacy

We first give the definition of Rényi differential privacy.

**Definition A.1** (($\alpha, \epsilon$)-Rényi DP [41]). *A data publication mechanism $(S, \pi)$ is $(\alpha, \epsilon)$-Rényi differentially private if for $(\theta, \theta') \in X$, we have*

$$\mathrm{D}_\alpha\left(\pi(\cdot|\theta)\|\pi(\cdot|\theta')\right) \le \epsilon,$$

*where $\mathrm{D}_\alpha(\pi(\cdot|\theta)\|\pi(\cdot|\theta'))$ is the $\alpha$-Rényi divergence between the two distributions $\pi(\cdot|\theta)$ and $\pi(\cdot|\theta')$.*

The definition of Rényi divergence is as follows.

**Definition A.2** (Rényi divergence [43]). *Let $P$ and $Q$ be probability distributions on $\Omega$. For $\alpha \in (1, \infty)$, we define the Rényi divergence of order $\alpha$ between $P$ and $Q$ as*

$$
\begin{aligned}
\mathrm{D}_\alpha(P\|Q) &= \frac{1}{\alpha - 1} \log\left(\int_\Omega P(x)^\alpha Q(x)^{1-\alpha} \mathrm{d}x\right) \\
&= \frac{1}{\alpha - 1} \log\left(\mathop{\mathbb{E}}_{x \sim Q}\left[\left(\frac{P(x)}{Q(x)}\right)^\alpha\right]\right) \\
&= \frac{1}{\alpha - 1} \log\left(\mathop{\mathbb{E}}_{x \sim P}\left[\left(\frac{P(x)}{Q(x)}\right)^{\alpha-1}\right]\right),
\end{aligned}
$$

*where $P(\cdot)$ and $Q(\cdot)$ are the probability mass/density functions of $P$ and $Q$.*

We then revise Program (1) and (2) in the form of Rényi differential privacy[7].

$$
\begin{aligned}
\max_{(S,\pi)} \quad & \sum_\theta \sum_s \pi(s|\theta)\mu(\theta)v(a_s, \theta) \\
s.t. \quad & \textstyle\sum_\theta u(a_s, \theta)\pi(s|\theta)\mu(\theta) \ge \sum_\theta u(a', \theta)\pi(s|\theta)\mu(\theta), \text{ for all } s \in S, a' \in A \\
& \textstyle\sum_{s \in S} \pi(s|\theta)\left(\frac{\pi(s|\theta)}{\pi(s|\theta')}\right)^{\alpha-1} \le e^{(\alpha-1)\epsilon}, && \text{for all } (\theta, \theta') \in X \\
& \textstyle\sum_s \pi(s|\theta) = 1, && \text{for all } \theta \in \Theta \\
& \pi(s|\theta) \ge 0, && \text{for all } \theta \in \Theta, s \in S
\end{aligned}
\tag{4}
$$

$$
\begin{aligned}
\max_\tau \quad & \mathbb{E}_{q_s \sim \tau}[V(q_s)] \\
s.t. \quad & \mathbb{E}_{q_s \sim \tau}[q_s(\theta)] = \mu(\theta), && \text{for all } \theta \in \Theta \\
& \mathbb{E}_{q_s \sim \tau}\left[\left(\frac{q_s(\theta)}{\mu(\theta)}\right)^\alpha \left(\frac{q_s(\theta')}{\mu(\theta')}\right)^{1-\alpha}\right] \le e^{(\alpha-1)\epsilon}, \text{ for all } (\theta, \theta') \in X
\end{aligned}
\tag{5}
$$

We now present a similar binary characterization as Proposition 3.1. Note that Proposition 3.2 also holds for Rényi differential privacy.

**Proposition A.3.** *We define $t_1 = q_{s_1}(1-\mu)/((1-q_{s_1})\mu)$, $t_2 = q_{s_2}(1-\mu)/((1-q_{s_2})\mu)$ and a distribution $\sigma$ with binary support on $\{t_1, t_2\}$. Also, $\sigma(t_1) = \frac{1-q_{s_1}}{1-\mu}\tau(q_{s_1})$ and $\sigma(t_2) = \frac{1-q_{s_2}}{1-\mu}\tau(q_{s_2})$. The signaling scheme preserves $(\alpha, \epsilon)$-Rényi DP if and only distribution $\sigma$ satisfy:*

$$
\begin{cases}
\mathbb{E}_\sigma[t] = 1 \\
\mathbb{E}_\sigma[t^\alpha] \le e^{(\alpha-1)\epsilon} \\
\mathbb{E}_\sigma[t^{1-\alpha}] \le e^{(\alpha-1)\epsilon}.
\end{cases}
$$

Proof. Recall the $(\alpha, \epsilon)$-Rényi DP constraint in the program using the form of posteriors:

$$
\mathbb{E}_{q_s \sim \tau}\left[\left(\frac{q_s(\theta)}{\mu(\theta)}\right)^\alpha \left(\frac{q_s(\theta')}{\mu(\theta')}\right)^{1-\alpha}\right] \le e^{(\alpha-1)\epsilon}, \text{ for all } (\theta, \theta') \in X.
$$

Then we obtain

$$
\left(\frac{q_{s_1}}{\mu}\right)^\alpha \left(\frac{1-\mu}{1-q_{s_1}}\right)^{\alpha-1} \tau(q_{s_1}) + \left(\frac{q_{s_2}}{\mu}\right)^\alpha \left(\frac{1-\mu}{1-q_{s_2}}\right)^{\alpha-1} \tau(q_{s_2}) \le e^{(\alpha-1)\epsilon},
$$

$$
\left(\frac{1-q_{s_1}}{1-\mu}\right)^\alpha \left(\frac{\mu}{q_{s_1}}\right)^{\alpha-1} \tau(q_{s_1}) + \left(\frac{1-q_{s_2}}{1-\mu}\right)^\alpha \left(\frac{\mu}{q_{s_2}}\right)^{\alpha-1} \tau(q_{s_2}) \le e^{(\alpha-1)\epsilon}.
$$

---

[7]For partial privacy, we substitute $X_M$ for $X$ and the results can be simply extended.

We then define $t_1, t_2, \sigma(t_1), \sigma(t_2)$ as the theorem. Substituting them into the above equations, we have

$$t_1^\alpha \sigma(t_1) + t_2^\alpha \sigma(t_2) \le e^{(\alpha-1)\epsilon},$$

$$t_1^{1-\alpha} \sigma(t_1) + t_2^{1-\alpha} \sigma(t_2) \le e^{(\alpha-1)\epsilon}.$$

Also, $t_1 \sigma(t_1) + t_2 \sigma(t_2) = (q_{s_1} \tau(q_{s_1}) + q_{s_2} \tau(q_{s_2}))/\mu = 1$.                                         □

For general characterization in Section 4, note that Rényi differential privacy can also be written in the form of Ex ante constraints. Then Lemma 4.5 implies only $k + m$ signals suffice. We then just modify the objective function $V$ similar to $(\epsilon, \delta)$-DP.

*A modified objective function for $(\alpha, \epsilon)$-Rényi DP.* Similarly, for $(\alpha, \epsilon)$-Rényi DP, we use the modified function $F(q, \gamma) = V(q)$ same as $(\epsilon, \delta)$-DP and define

$$C'(\mu, \alpha) := \left\{ (q, \gamma) \in \Delta(\Theta) \times \mathbb{R}^m : \left(\frac{q(\theta)}{\mu(\theta)}\right)^\alpha \left(\frac{q(\theta')}{\mu(\theta')}\right)^{1-\alpha} \le \gamma_{\theta,\theta'}, \text{ any } (\theta, \theta') \in X \right\}.$$

We then characterize the problem with the concave hull of $F$.

**Proposition A.4.** *For $(\alpha, \epsilon)$-Rényi DP, there exists a valid optimal signaling scheme such that*
- *The scheme uses $|S_F(\mu, e^{(\alpha-1)\epsilon} \mid C'(\mu, \alpha))|$ signals, which is not larger than $k + m$. Here $e^{(\alpha-1)\epsilon}$ is the $m$-dimension vector with all elements $e^{(\alpha-1)\epsilon}$.*
- *The scheme induces posteriors in $S_F(\mu, e^{(\alpha-1)\epsilon} \mid C'(\mu, \alpha))$.*
- *The optimal utility of the sender is $\overline{F}(\mu, e^{(\alpha-1)\epsilon} \mid C'(\mu, \alpha))$.*

**Corollary A.5.** *Sender benefits from persuasion under $(\alpha, \epsilon)$-Rényi DP if and only if $\overline{F}(\mu, e^{(\alpha-1)\epsilon} \mid C'(\mu, \alpha)) > V(\mu)$.*

# B  More Analyses about Beneficial Persuasion Conditions

We first relate the beneficial persuasion condition to the concavity/convexity of the objective function over the feasible region.

**Proposition B.1.** *Under $\epsilon$-DP, if $V$ is concave in $K(\mu, \epsilon)$, the sender does not benefit from persuasion for any prior. If $V$ is convex and not concave in $K(\mu, \epsilon)$, the sender benefits from persuasion for any prior.*

**Proposition B.2.** *Under $(\epsilon, \delta)$-DP, if $F$ is concave in $C(\mu, \epsilon)$, the sender does not benefit from persuasion for any prior. If $F$ is convex and not concave in $C(\mu, \epsilon)$, the sender benefits from persuasion for any prior.*

**Proposition B.3.** *Under $(\alpha, \epsilon)$-Rényi DP, if $F$ is concave in $C'(\mu, \alpha)$, the sender does not benefit from persuasion for any prior. If $F$ is convex and not concave in $C'(\mu, \alpha)$, the sender benefits from persuasion for any prior.*

When the objective function is neither concave nor convex, alternative techniques can determine whether the sender benefits from persuasion. Intuitively, the sender can gain from persuasion if they can sometimes persuade the receiver to switch from their default action to one the sender prefers instead. The receiver's default action without knowing more information from the sender is $\hat{a}(\mu)$. When the posterior is $q$, the sender's utility without persuasion can be seen as $\mathbb{E}_{\theta \sim q}[v(\hat{a}(\mu), \theta)]$. Also, if the sender shares the posterior with the receiver, the utility is $V(q)$. Therefore, we define the condition "the sender is willing to share information" for different privacy constraints here.

Say "the sender is willing to share information" under $\epsilon$-DP if there exists $q \in K(\mu, \epsilon)$ that

$$V(q) > \mathbb{E}_{\theta \sim q}[v(\hat{a}(\mu), \theta)].$$

Say "the sender is willing to share information" under $(\epsilon, \delta)$-DP if there exists $(q, \gamma) \in C(\mu, \epsilon)$ that

$$F(q, \gamma) = V(q) > \mathbb{E}_{\theta \sim q}[v(\hat{a}(\mu), \theta)].$$

Say "the sender is willing to share information" under $(\alpha, \epsilon)$-Rényi DP if there exists $(q, \gamma) \in C'(\mu, \alpha)$ that

$$F(q, \gamma) = V(q) > \mathbb{E}_{\theta \sim q}[v(\hat{a}(\mu), \theta)].$$

**Proposition B.4.** *If "the sender is willing to share information" doesn't hold, the sender cannot benefit from persuasion.*

An additional assumption is helpful to definitively state the condition under which the sender benefits from persuasion. Beyond requiring the existence of a posterior the sender is willing to share, it is also useful to assume the receiver's optimal action does not change in a neighborhood around the prior belief.

Formally, we say the receiver's preference is discrete at $\mu$ if there exists $\epsilon > 0$ such that for any $a \ne \hat{a}(\mu)$, $\mathbb{E}_\mu[u(\hat{a}(\mu), \theta)] > \mathbb{E}_\mu[u(a, \theta) + \epsilon]$. When the action space $A$ is finite, the receiver's preference is non-discrete only if they are indifferent between two actions, which occurs at finite $\mu$. Thus, this assumption is fairly mild for finite action spaces.

**Assumption B.5.** *The receiver's preference is discrete at the prior $\mu$.*

**Proposition B.6.** *Under Assumption B.5, if "the sender is willing to share information" holds, the sender benefits from the persuasion.*

## C Missing Proofs in Section 3

### C.1 Proof of Proposition 3.1

For the first point that two signals suffice, we delay the proof to Lemma D.1. Recall the $\epsilon$-DP constraint in the program using the form of posteriors:

$$\frac{q_s(\theta)\mu(\theta')}{q_s(\theta')\mu(\theta)} \le e^\epsilon, \text{ for all } q_s \text{ and } (\theta, \theta') \in X.$$

Therefore, for every signal $s$ in the signal scheme, its posterior $q_s$ satisfies

$$q_s - q_s\mu \le e^\epsilon \mu(1 - q_s),$$
$$\mu - \mu q_s \le e^\epsilon(q_s - q_s\mu).$$

The results can be obtained by slightly organizing the above two equations.

Recall the $(\epsilon, \delta)$-DP constraint in the program using the form of posteriors:

$$\sum_{q_s \in Q} \frac{q_s(\theta)\tau(q_s)}{\mu(\theta)} \le e^\epsilon \sum_{q_s \in Q} \frac{q_s(\theta')\tau(q_s)}{\mu(\theta)} + \delta, \text{ for all } Q \subset supp(\tau) \text{ and } (\theta, \theta') \in X.$$

Since $q_{s_2} \ge q_{s_1}$ and $\mathbb{E}[q_s] = \mu$, we have $q_{s_2} \ge \mu \ge q_{s_1}$. Also, $\tau(q_{s_1}) = (q_{s_2} - \mu)/(q_{s_2} - q_{s_1})$ and $\tau(q_{s_2}) = (\mu - q_{s_1})/(q_{s_2} - q_{s_1})$. Therefore, we can simplify the above inequations to

$$\left(\frac{1 - q_{s_1}}{1 - \mu} - e^\epsilon \frac{q_{s_1}}{\mu}\right) \frac{q_{s_2} - \mu}{q_{s_2} - q_{s_1}} \le \delta,$$
$$\left(\frac{q_{s_2}}{\mu} - e^\epsilon \frac{1 - q_{s_2}}{1 - \mu}\right) \frac{\mu - q_{s_1}}{q_{s_2} - q_{s_1}} \le \delta,$$

which implies the result by slight organization.

### C.2 Proof of Theorem 3.4

We first choose $\mu < t$ that satisfy

$$\frac{\mu}{e^{-\epsilon_1} - e^{-\epsilon_1}\mu + \mu} + \frac{(1 - C)\mu(1 - \mu)}{e\mu + 1 - \mu}\delta = t.$$

Since when $\mu = 0$, the left side of the above equation is smaller than the right side, and when $\mu = t$, the right side is larger than the right side. Also, the left side is continuous for $\mu$ between 0 and $t$, and the right side is constant. Therefore, there exists a $\mu \in (0, t)$ that satisfies the equation.

With this construction, under $(\epsilon_1, 0)$-differential privacy, the sender cannot get positive utility under any signal scheme.

Then we try to design a good signal scheme under $(\epsilon_2, \delta)$-differential privacy. Let one posterior $q_{s_2}$ be $t$ and another posterior $q_{s_1}$ will be assigned later. Then for $(\theta = 0, \theta' = 1)$, the constraint is

$$\left(\frac{q_{s_2}}{\mu} - e^{\epsilon_2} \frac{1 - q_{s_2}}{1 - \mu}\right) \frac{\mu - q_{s_1}}{q_{s_2} - q_{s_1}} \le \left(\frac{1 - e^{\epsilon_2 - \epsilon_1}}{e^{-\epsilon_1}(1 - \mu) + \mu} + (1 - C)\delta\right) \frac{\mu - q_{s_1}}{q_{s_2} - q_{s_1}} \le \delta$$

and for $(\theta = 1, \theta' = 0)$, the constraint is

$$\left(\frac{1 - q_{s_1}}{1 - \mu} - e^{\epsilon_2} \frac{q_{s_1}}{\mu}\right) \frac{q_{s_2} - \mu}{q_{s_2} - q_{s_1}} \le \delta.$$

We now assign $q_{s_1}$ as $\mu/(e^{\epsilon_2}(1 - \mu) + \mu)$, then the second inequality is naturally satisfied. For the first inequality,

$$\left(\frac{1 - e^{\epsilon_2 - \epsilon_1}}{e^{-\epsilon_1}(1 - \mu) + \mu} + (1 - C)\delta\right) \frac{\mu - q_{s_1}}{q_{s_2} - q_{s_1}} \overset{(a)}{\le} \frac{C\delta}{e^{-\epsilon_1}(1 - \mu) + \mu} \cdot \frac{\mu - q_{s_1}}{q_{s_2} - q_{s_1}} + (1 - C)\delta$$

$$\overset{(b)}{\le} \frac{C\delta}{e^{-\epsilon_1}(1 - \mu) + \mu} \cdot \frac{1 - \frac{1}{e^{\epsilon_1}(1 - \mu) + \mu}}{\frac{1}{e^{-\epsilon_1}(1 - \mu) + \mu} - \frac{1}{e^{\epsilon_1}(1 - \mu) + \mu}} + (1 - C)\delta$$

$$= \frac{C\delta(e^{\epsilon_1} - 1)}{e^{\epsilon_1} - e^{-\epsilon_1}} + (1 - C)\delta$$

$$\le \delta.$$

In the above set of derivations, (a) holds since $\epsilon_1 - \epsilon_2 \le C\delta$ and $e^x \ge 1 + x$ for any $x$. (b) holds because $(\mu - q_{s_1})/(q_{s_2} - q_{s_1})$ increases when $q_{s_1}$ decreases and $q_{s_2}$ increases.

Therefore, $q_{s_1} = \mu/(e^{\epsilon_2}(1-\mu) + \mu)$ and $q_{s_2} = t$ is a feasible signal scheme under $(\epsilon_2, \delta)$-differential privacy. Using this signal scheme, the sender can at least get the utility of

$$\frac{\mu - q_{s_1}}{q_{s_2} - q_{s_1}} \overset{(a)}{\geq} \frac{(e^{\epsilon_2} - 1)(e^{-\epsilon_1}(1-\mu) + \mu)}{(e^{\epsilon_2} - e^{-\epsilon_1}) + \delta(e^{-\epsilon_1}(1-\mu) + \mu)(e^{\epsilon_2}(1-\mu) + \mu)}$$

$$\overset{(b)}{\geq} \frac{e^{\epsilon_2} - 1}{(1+\delta)e^{\epsilon_1+\epsilon_2} - 1}.$$

(a) holds due to $0 < 1 - C \leq 1$. (b) is obtained from $e^{-\epsilon_1} \leq e^{-\epsilon_1}(1-\mu) + \mu \leq 1$ and $e^{\epsilon_2}(1-\mu) + \mu \leq e^{\epsilon_2}$.

## C.3   Proof of Proposition 3.6

We now choose

$$\mu = \frac{te^{-\epsilon_1}}{1 + \omega - t + te^{-\epsilon_1}},$$

where $\omega$ is an arbitrarily small positive number. Then we obtain

$$\frac{(1+\omega)\mu}{e^{-\epsilon_1} - e^{-\epsilon_1}\mu + \mu} = t.$$

Using this construction, under $(\epsilon_1, 0)$-differential privacy, the sender cannot get positive utility under any signal scheme.

Then we try to design a good signal scheme under $(\epsilon_2, \delta)$-differential privacy. Let one posterior $q_{s_2}$ be 1 and another posterior $q_{s_1}$ will be assigned later. Then for $(\theta = 0, \theta' = 1)$, the constraint is

$$q_{s_1} \geq \frac{\mu(1-\delta)}{(1 - \delta\mu)},$$

and for $(\theta = 1, \theta' = 0)$, the constraint is

$$q_{s_1} \geq \frac{\mu(1-\delta)}{(e^{\epsilon_2} - \mu e^{\epsilon_2} + \mu - \delta\mu)},$$

which can both be satisfied when $q_{s_1} = \mu(1-\delta)/(1 - \delta\mu) < \mu$.

Therefore, under $(\epsilon_2, \delta)$-differential privacy, there exists a signal scheme to help the sender get positive utility, which implies the result.

## C.4   Proof of Proposition 3.7

Similar as the proof of Proposition 3.6, we set $\mu = te^{-\epsilon}/(1 + \omega - t + te^{-\epsilon})$, the sender then can not get positive utility under $\epsilon$-DP.

However, without the privacy constraint, the sender can let one posterior be 0 and another be $t$, which gives the utility of

$$\frac{\mu}{t} = \frac{1}{(1 + \omega - t)e^{\epsilon} + t}$$

. Since $\omega$ can be arbitrarily small, we then prove the result.

## C.5   Proof of Proposition 3.9

We now choose $\mu = t/2$.

Under $(\epsilon, \delta)$-differential privacy, the best choice of $q_{s_2}$ should lie in a region with utility 1 and at the same time have the largest weight in the expected utility, which is $t = 2\mu$.

Then for $(\theta = 0, \theta' = 1)$, the constraint is

$$\left(\frac{q_{s_2}}{\mu} - e^{\epsilon}\frac{1 - q_{s_2}}{1 - \mu}\right)\frac{\mu - q_{s_1}}{q_{s_2} - q_{s_1}} = \left(2 - e^{\epsilon}\frac{1 - 2\mu}{1 - \mu}\right)\frac{\mu - q_{s_1}}{2\mu - q_{s_1}} \leq \delta$$

and for $(\theta = 1, \theta' = 0)$, the constraint is

$$\left(\frac{1 - q_{s_1}}{1 - \mu} - e^{\epsilon}\frac{q_{s_1}}{\mu}\right)\frac{\mu}{2\mu - q_{s_1}} \leq \delta.$$

By reorganizing the above two inequalities, we obtain that the distance d between another posterior and the prior should satisfy:

$$d \leq \frac{\mu(1-\mu)\delta}{\max\{(1-\mu)\delta, (2 - e^{\epsilon} - \delta)(1-\mu) + e^{\epsilon}\mu\}}$$

and

$$d \leq \frac{(1-\mu)\mu(\delta + e^{\epsilon} - 1)}{\mu + (e^{\epsilon} - \delta)(1-\mu)}.$$

So the utility of the sender should be less than $\frac{d}{\mu+d}$.

However, without the privacy constraint, the sender can let one posterior be 0 and another be $2\mu$, which gives a utility of $1/2$ and implies the proposition.

# D Missing Proofs in Section 5

We first show that using the same number of signals as the number of actions is enough to give an optimal signaling scheme.

**Lemma D.1.** *There always exists an optimal signaling scheme that uses at most signals as the number of receiver actions.*

PROOF. If signal $s$ and $s'$ result in the same optimal action profile $a$, Sender can instead send a new signal $s_a = (s, s')$ in both cases. Therefore, we have $\pi(s_a|\theta) = \pi(s|\theta) + \pi(s'|\theta)$ for any $\theta$. Merging signals doesn't affect the value of objective function $\sum_\theta \sum_s \pi(s|\theta)\mu(\theta)v(a_s, \theta)$. We then show that merging signals also does not break the constraints that already hold.

Bringing this equation back into the above programming, privacy constraints for $(\epsilon, \delta)$-DP will not be violated since $\max\{0, \pi(s|\theta) - e^\epsilon \pi(s|\theta')\} + \max\{0, \pi(s'|\theta) - e^\epsilon \pi(s'|\theta')\} \geq \max\{0, \pi(s_a|\theta) - e^\epsilon \pi(s_s|\theta')\}$ holds for all $(\theta, \theta') \in X$. □

Then $2^t$ signals suffice, corresponding to selecting a subset of receivers to suggest action 1, with others suggested action 0. We slightly overload notation, also using $T \subseteq \{1, \ldots, t\}$ to denote the signal targeting subset $T$ for action 1.

We can write similar linear programming for this setting[8].

$$
\begin{aligned}
\max_\pi \quad & \sum_\theta \sum_T \mu(\theta)v(T, \theta)\pi(T|\theta) \\
s.t. \quad & \sum_\theta \left(\mu(\theta)(u_i(1, \theta) - u_i(0, \theta)) \sum_{T:i\in T} \pi(T|\theta)\right) \geq 0, \text{ for all } i \in \{1, \cdots, t\} \\
& \sum_\theta \left(\mu(\theta)(u_i(0, \theta) - u_i(1, \theta)) \sum_{T:i\notin T} \pi(T|\theta)\right) \geq 0, \text{ for all } i \in \{1, \cdots, t\} \\
& \sum_{T:i\in T} \pi(T|\theta) \leq e^\epsilon \sum_{T:i\in T} \pi(T|\theta') + \delta, \quad \text{for all } i \text{ and } (\theta, \theta') \in X \\
& \sum_{T:i\notin T} \pi(T|\theta) \leq e^\epsilon \sum_{T:i\notin T} \pi(T|\theta') + \delta, \quad \text{for all } i \text{ and } (\theta, \theta') \in X \\
& \sum_T \pi(T|\theta) = 1, \quad \text{for all } \theta \\
& \pi(T|\theta) \geq 0, \quad \text{for all } \theta, T
\end{aligned} \quad (6)
$$

For the oblivious method, the signal distribution depends only on the projection $\omega = \phi(\theta)$. We can then redefine adjacency under this oblivious scheme. Say $\omega, \omega'$ adjacent if and only if $|\omega - \omega'| = 1$. Let $X^O$ denote the set of all adjacent pairs $(\omega, \omega')$ here[9]. Under the oblivious method, Program (6) can be modified into:

$$
\begin{aligned}
\max_\pi \quad & \sum_\omega \sum_T \mu(\omega)v(T, \omega)\pi(T|\omega) \\
s.t. \quad & \sum_\omega \left(\mu(\omega)(u_i(1, \omega) - u_i(0, \omega)) \sum_{T:i\in T} \pi(T|\omega)\right) \geq 0, \text{ for all } i \in \{1, \cdots, t\} \\
& \sum_\omega \left(\mu(\omega)(u_i(0, \omega) - u_i(1, \omega)) \sum_{T:i\notin T} \pi(T|\omega)\right) \geq 0, \text{ for all } i \in \{1, \cdots, t\} \\
& \sum_{T:i\in T} \pi(T|\omega) \leq e^\epsilon \sum_{T:i\in T} \pi(T|\omega') + \delta, \quad \text{for all } i \text{ and } (\omega, \omega') \in X^O \\
& \sum_{T:i\notin T} \pi(T|\omega) \leq e^\epsilon \sum_{T:i\notin T} \pi(T|\omega') + \delta, \quad \text{for all } i \text{ and } (\omega, \omega') \in X^O \\
& \sum_T \pi(T|\omega) = 1, \quad \text{for all } \omega \\
& \pi(T|\omega) \geq 0, \quad \text{for all } \omega, T
\end{aligned} \quad (7)
$$

We also prove that the simplified version is equivalent to the original one, which means the oblivious method is without loss.

**Lemma D.2.** *Under Assumption 5.1, Program (6) and Program (7) are equivalent. Therefore, there exists an oblivious privacy-preserving signaling scheme that is optimal among all privacy-preserving signaling schemes.*

For the simplicity of presentation, we delay the proof of Lemma D.2 to Appendix D.1.

Consider the following dual program with variables $\alpha_i^1, \alpha_i^0, \beta_{i,(\omega,\omega')}, \gamma_{i,(\omega,\omega')}, y_\omega$.

$$
\begin{aligned}
\min \quad & \sum_\omega y_\omega + \delta \sum_{i,\omega} (\beta_{i,(\omega,\omega')} + \gamma_{i,(\omega,\omega')}) \\
s.t. \quad & -\sum_{i:i\in T} \mu(\omega)(u_i(1, \omega) - u_i(0, \omega))\alpha_i^1 - \sum_{i:i\notin T} \mu(\omega)(u_i(0, \omega) - u_i(1, \omega))\alpha_i^0 \\
& + \sum_{i:i\in T} \sum_{\omega':(\omega,\omega')\in X^O} (\beta_{i,(\omega,\omega')} - e^\epsilon \beta_{i,(\omega',\omega)}) \\
& + \sum_{i:i\notin T} \sum_{\omega':(\omega,\omega')\in X^O} (\gamma_{i,(\omega,\omega')} - e^\epsilon \gamma_{i,(\omega',\omega)}) + y_\omega \geq \mu(\omega)v(T, \omega), \text{ for all } \omega, T \\
& \alpha_i^1, \alpha_i^0, \beta_{i,(\omega,\omega')}, \gamma_{i,(\omega,\omega')} \geq 0, \text{ for all } i \text{ and } (\omega, \omega') \in X^O
\end{aligned} \quad (8)
$$

We can obtain a separation oracle for the program given an algorithm discussed later. Given any variables $\alpha_i^1, \alpha_i^0, \beta_{i,(\omega,\omega')}, \gamma_{i,(\omega,\omega')}, y_\omega$, separation over the first set of constraints reduces to maximizing the set function

$$
\begin{aligned}
g_\omega(T) = & v(T, \omega) + \frac{1}{\mu(\omega)} \Bigg( \sum_{i:i\in T} \mu(\omega)(u_i(1, \omega) - u_i(0, \omega))\alpha_i^1 + \sum_{i:i\notin T} \mu(\omega)(u_i(0, \omega) - u_i(1, \omega))\alpha_i^0 \\
& - \sum_{i:i\in T} \sum_{\omega':(\omega,\omega')\in X^O} (\beta_{i,(\omega,\omega')} - e^\epsilon \beta_{i,(\omega',\omega)}) - \sum_{i:i\notin T} \sum_{\omega':(\omega,\omega')\in X^O} (\gamma_{i,(\omega,\omega')} - e^\epsilon \gamma_{i,(\omega',\omega)}) \Bigg)
\end{aligned}
$$

---

[8]For partial privacy, we replace $X$ with $X_M$.

[9]For partial privacy, say $\omega = (a, b), \omega' = (a', b')$ adjacent if $|a - a'| = 1$ and $b = b'$. Let $X_M^O$ denote the set of all adjacent pairs. We substitute $X_M^O$ for $X^O$.

for each $\omega$. The other constraints can be checked directly in linear time. Given the resulting separation oracle, we can use the Ellipsoid method to obtain a vertex optimal solution to both Program (7) and its dual Program (8) in polynomial time.

We reorganize the set function $g$ as

$$g_\omega(T) = v(T, \omega) + \frac{1}{\mu(\omega)} \sum_{i:i\in T} \Big( \mu(\omega)(u_i(1,\omega) - u_i(0,\omega))\alpha_i^1 - \mu(\omega)(u_i(0,\omega) - u_i(1,\omega))\alpha_i^0 - \sum_{\omega':(\omega,\omega')\in X^O} (\beta_{i,(\omega,\omega')} - e^\epsilon \beta_{i,(\omega',\omega)}) +$$

$$\sum_{\omega':(\omega,\omega')\in X^O} (\gamma_{i,(\omega,\omega')} - e^\epsilon \gamma_{i,(\omega',\omega)}) \Big) + \frac{1}{\mu(\omega)} \sum_i \Big( \mu(\omega)(u_i(0,\omega) - u_i(1,\omega))\alpha_i^0 - \sum_{\omega':(\omega,\omega')\in X^O} (\gamma_{i,(\omega,\omega')} - e^\epsilon \gamma_{i,(\omega',\omega)}) \Big).$$

Therefore, for any $\omega$, there is a polynomial time algorithm to find $T$ that maximizes $g_\omega(T)$. For every $|T| \in \{0, 1, \cdots, t\}$, the optimal set is to pick the $|T|$ biggest elements in $w_i$, $i \in \{1, \cdots, t\}$, where $w_i$ is

$$\mu(\omega)(u_i(1,\omega) - u_i(0,\omega))\alpha_i^1 - \mu(\omega)(u_i(0,\omega) - u_i(1,\omega))\alpha_i^0 - \sum_{\omega':(\omega,\omega')\in X^O} (\beta_{i,(\omega,\omega')} - e^\epsilon \beta_{i,(\omega',\omega)}) + \sum_{\omega':(\omega,\omega')\in X^O} (\gamma_{i,(\omega,\omega')} - e^\epsilon \gamma_{i,(\omega',\omega)})$$

Each $w_i$ can be computed in polynomial time. Enumerating all $|T|$ yields the optimal set $T$. Therefore, we obtain Theorem 5.2.

## D.1 Proof of Lemma D.2

For generalization of the proof, we present it with the form of partial privacy, including $M$ to denote the set of bits whose privacy should be considered. For the original case, we only need to consider $M = \{1, 2, \cdots, n\}$ and replace $X_M$ with $X$, $\phi_M$ with $\phi$ and $X_M^O$ with $X^O$.

We first notice that any feasible oblivious solution for Program (7) is also a feasible solution for Program (6).

**Lemma D.3.** *Under Assumption 5.1, for all kinds of privacy constraints, if an oblivious signaling scheme satisfies Program (7), we can construct the following signaling scheme:*

$$\pi'(T|\theta) = \pi(T|\phi_M(\theta))$$

*which satisfies the original program.*

Proof. We first show that the new signaling scheme satisfies all basic constraints in the program. For all $\theta$, from assumptions, we have

$$\sum_\theta \left( \mu(\theta)(u_i(1,\theta) - u_i(0,\theta)) \sum_{T:i\in T} \pi'(T|\theta) \right) = \sum_\omega \left( \sum_{\theta:\phi_M(\theta)=\omega} \mu(\theta)(u_i(1,\omega) - u_i(0,\omega)) \sum_{T:i\in T} \pi(T|\omega) \right)$$

$$= \sum_\omega \left( \mu(\omega)(u_i(1,\omega) - u_i(0,\omega)) \sum_{T:i\in T} \pi(T|\omega) \right)$$

$$\geq 0, \text{ for all } i \in \{1, \cdots, t\}.$$

The other side is similar. $\sum_T \pi'(T|\theta) = 1$ and $\pi'(T|\theta) \geq 0$ holds naturally.

Also, for $(\theta, \theta') \in X_M$, $(\phi_M(\theta), \phi_M(\theta')) \in X_M^O$. Therefore, $\pi'(T|\theta)$ satisfies the privacy constraints in the original program, which implies the lemma. □

Let $F_0$ be the set of feasible signaling schemes for Program (6). An oblivious scheme can be constructed by projecting any signaling scheme in $F_0$. For signaling scheme in $F_0$, we can build an oblivious signaling scheme through projection:

$$\pi'(T|\omega) = \frac{1}{\mu(\omega)} \sum_{\theta:\phi_M(\theta)=\omega} \pi(T|\theta)\mu(\theta).$$

Let $PF_0$ be the set of projections of feasible signaling schemes to oblivious schemes.

Also, let $F_1$ be the set of feasible oblivious signaling schemes for Program (7). Note that in Lemma D.3, the feasible oblivious signaling scheme can be seen as the projection of the constructed signaling scheme, then it shows that $F_1 \subseteq PF_0$. We then show that $PF_0 \subseteq F_1$ as well.

**Lemma D.4.** *Under Assumption 5.1, any projection of a feasible signaling scheme is also a feasible oblivious signaling scheme.*

Proof. Consider any signaling scheme in $F_0$, the projection is $\pi'(T|\omega) = \frac{1}{\mu(\omega)} \sum_{\theta:\phi_M(\theta)=\omega} \pi(T|\theta)\mu(\theta)$.

We now show that $\pi'(T|\omega)$ is a feasible oblivious signaling scheme. First, $\sum_T \pi'(T|\omega) = 1$ holds for all $\omega$. $\pi'(T|\omega) \geq 0$ holds for all $\omega$ and $T$. Also,

$$\sum_\omega \left( \mu(\omega)(u_i(1,\omega) - u_i(0,\omega)) \sum_{T:i\in T} \pi'(T|\omega) \right) = \sum_\theta \left( \mu(\theta)(u_i(1,\theta) - u_i(0,\theta)) \sum_{T:i\in T} \pi(T|\theta) \right) \geq 0, \text{ for all } i \in \{1, \cdots, t\}.$$

The other side is similar. We then show that it still holds the privacy constraint. Use $m$ to denote the size of $M$, $\omega_1$ to denote the first number in $\omega$ and $\omega_2$ to denote the second number. Notice that for any $\omega = (\omega_1, \omega_2)$ that $\omega_1 \in \{0, 1, \cdots, m-1\}$, considering its adjacent state $\omega' = (\omega_1 + 1, \omega_2)$,

$$\pi'(T|\omega) = \frac{1}{\binom{m}{\omega_1}\binom{n-m}{\omega_2}(m-\omega_1)} \sum_{\theta:\phi_M(\theta)=\omega} \sum_{i:\theta_i=0, i\in M} \pi(T|\theta) \tag{9}$$

and

$$\pi'(T|\omega') = \frac{1}{\binom{m}{\omega_1+1}\binom{n-m}{\omega_2}(\omega_1+1)} \sum_{\theta:\phi_M(\theta)=\omega} \sum_{i:\theta_i=0, i\in M} \pi(T|(1,\theta_{-i})) = \frac{1}{\binom{m}{\omega_1}\binom{n-m}{\omega_2}(m-\omega_1)} \sum_{\theta:\phi_M(\theta)=\omega} \sum_{i:\theta_i=0, i\in M} \pi(T|(1,\theta_{-i})). \tag{10}$$

For $(\epsilon, \delta)$-DP, since for any $\omega = (\omega_1, \omega_2)$ that $\omega_1 \in \{0, 1, \cdots, m-1\}$, considering its adjacent state $\omega' = (\omega_1 + 1, \omega_2)$, for all $\theta$ that $\phi_M(\theta) = \omega$ and all $i$ that $\theta_i = 0$,

$$\sum_{T:i\in T} \left( \pi(T|\theta) - e^\epsilon \pi(T|(1,\theta_{-i})) \right) \leq \delta,$$

we then have

$$\sum_{T:i\in T} \left( \pi'(T|\omega) - e^\epsilon \pi(T|\omega') \right) = \sum_{T:i\in T} \left( \frac{1}{\binom{m}{\omega_1}\binom{n-m}{\omega_2}(m-\omega_1)} \sum_{\theta:\phi_M(\theta)=\omega} \sum_{i:\theta_i=0, i\in M} \left( \pi(T|\theta) - e^\epsilon \pi(T|(1,\theta_{-i})) \right) \right)$$

$$\leq \frac{1}{\binom{m}{\omega_1}\binom{n-m}{\omega_2}(m-\omega_1)} \sum_{\theta:\phi_M(\theta)=\omega} \sum_{i:\theta_i=0, i\in M} \sum_{T:i\in T} \left( \pi(T|\theta) - e^\epsilon \pi(T|(1,\theta_{-i})) \right)$$

$$\leq \delta.$$

Similarly, the other side holds.

$\square$

**Lemma D.5.** *Under Assumption 5.1, there exists an optimal signaling scheme in $PF_0$ for Program (6).*

PROOF. Considering the optimal signaling scheme in Program (6), we now project it and obtain the scheme that

$$\pi'(T|\theta) = \frac{1}{\sum_{\theta':\phi_M(\theta')=\phi_M(\theta)} \mu(\theta')} \sum_{\theta':\phi_M(\theta')=\phi_M(\theta)} \pi(T|\theta')\mu(\theta').$$

We then obtain

$$\sum_{\theta\in\Theta} \sum_T \pi'(T|\theta)\mu(\theta)v(T,\theta) = \sum_{\theta\in\Theta} \sum_T \pi(T|\theta)\mu(\theta)v(T,\theta),$$

which means the signaling scheme achieves the same sender's utility as the original optimal one.

$\square$

From all the lemmas above, we prove that Program (6) and Program (7) are equivalent.

