# OpenReview forum: "Differentially Private Bayesian Persuasion"
_ACM.org/TheWebConf/2025/Conference — WWW 2025 Oral_

### Official Review · Reviewer_icSi · 2024-11-27

**Novelty:** 4
**Technical Quality:** 7

**Review:**

The authors study the tension between persuasion and privacy preservation.
They study persuasion in the framework of Bayesian persuasion and privacy preservation in the framework of differential privacy.
They explore two perspectives within Bayesian persuasion to understand how privacy constraints affect information disclosure.

The authors make three main contributions:
1. They that there is a constant utility gap in a sensible family of  utility functions.
2. They geometrically characterize the optimal signaling schemes for a number of popular privacy constraints.
3. They develop polynomial-time algorithms for computing optimal signaling schemes.

Pros:
- The claims are well supported by the technical contributions and analyses.
- The authors prominent notions of privacy.
- The authors shed light on the relationship between 2 important notions of privacy by studying the cost of privacy relative to each other.
- The authors generalize their results to a partial privacy setting.
- The paper is clear and well-written.

Cons:
- The motivating settings and examples could be developed earlier to aid intuition.
- Some of the simplifying assumptions seem to reduce the persuasion setting to, or at least significantly towards, the differential privacy setting (e.g. when the sender's utility is the sum of the receviers' utilities, and their utilities are independent of each other). The paper does not clearly lay out how the simplified setting continues to be different from standard differential privacy and what insights the simplified setting yields.

**Questions:**

Around line 276, the authors state that in the main part, the sender's utility is the sum of the receviers' utilities (line 276).
Doesn't this sort of simplify away the tension between the sender and receivers in the Bayesian persuasion problem?
In other words, when the sender's utility is the same as the receiver's utilities and the receviers' utilities are independent of each other and the sender can signal each independently, then how does this significantly differ from the standard differential privacy setting?

**Reviewer Confidence:**

2: The reviewer is willing to defend the evaluation, but it is likely that the reviewer did not understand parts of the paper

**Scope:**

3: The work is somewhat relevant to the Web and to the track, and is of narrow interest to a sub-community

---

### Official Review · Reviewer_ef3v · 2024-11-29

**Novelty:** 5
**Technical Quality:** 5

**Review:**

*Summary:*

This paper explores the problem of Bayesian persuasion under differential privacy constraints. The authors first analyze the scenario involving one receiver and one client, then extend the study to more general settings with partial privacy guarantees and multiple receivers.

"Strengths:"
1. The presentation is clear and easy to follow, making the theoretical analysis accessible.
2. The authors provide a thorough theoretical analysis of three types of differential privacy guarantees: $\epsilon$-DP, $(\epsilon, \delta)$-DP, and Rényi DP.

"Weakness:"
1. The paper would benefit from discussing more related work on Bayesian persuasion, particularly providing additional explanations on how these concepts can be applied in practical scenarios.

**Questions:**

See weakness

**Reviewer Confidence:**

1: The reviewer's evaluation is an educated guess

**Scope:**

2: The connection to the Web is incidental, e.g., use of Web data or API

---

### Official Review · Reviewer_tM1U · 2024-12-01

**Novelty:** 6
**Technical Quality:** 6

**Review:**

The paper is of high quality, presenting a robust theoretical framework for Bayesian persuasion under differential privacy constraints. It effectively balances the need for privacy with the utility of information dissemination. The clarity of exposition is commendable, with complex concepts and mathematical proofs presented in a structured and understandable manner. The use of formal mathematical language and well-organized sections facilitates comprehension and highlights the rigor of the research.

This work is original in its integration of differential privacy into the Bayesian persuasion framework. The approach of modifying the signaling scheme to accommodate various levels of privacy constraints (ε-differential privacy, (ε, δ)-differential privacy, and Rényi differential privacy) is novel and contributes significantly to the literature on privacy-preserving mechanisms in economic theory and game theory.

The significance of this paper lies in its potential impact on fields where information needs to be shared but privacy must be maintained, such as online advertising, healthcare data sharing, and personalized recommendations. The mathematical results and algorithms provided could significantly influence future research and applications in these areas, offering a method to optimize information dissemination while strictly controlling privacy leakage.

Pros
Theoretical Depth: The paper offers deep theoretical insights combined with practical algorithms that can be applied in real-world scenarios, which enhances its utility and appeal.
Methodological Innovation: The method of adapting signaling schemes under different privacy constraints represents a significant advancement in the field.

Cons:
Implementation Details: While the paper provides algorithms for computing optimal signaling schemes, it lacks detailed discussion on their computational efficiency in practice, especially for large-scale applications.
Assumption Sensitivity: The models rely on specific assumptions about privacy parameters and utility functions, which might not hold in all practical scenarios, potentially limiting the generalizability of the findings.

**Questions:**

1. What are the scalability challenges of your proposed algorithms? Are there specific strategies you recommend for handling the exponential growth in variables when the number of agents or complexity of the data increases?

**Reviewer Confidence:**

3: The reviewer is confident but not certain that the evaluation is correct

**Scope:**

3: The work is somewhat relevant to the Web and to the track, and is of narrow interest to a sub-community

---

### Official Review · Reviewer_44SX · 2024-12-02

**Novelty:** 3
**Technical Quality:** 4

**Review:**

Summary: The paper combines the paradigm of bayesian persuation with differential privacy. In bayesian persuasion an informed sender needs to persuade a less informed receiver through some signal to take an action that will benefit the sender. The paper combines this idea with differential privacy, which ensures that the signals send will not compromise the privacy of individuals.

Pros:

* The paper is on scope of the conference and the problem is interesting.
* The paper provides non-trivial results

Cons:

* The paper is not particularly well written and is quite hard to follow
* The contribution of the paper is not clear.

Some detailed comments:

1. My most important concern is that the writing of the paper lacks engagement and does not clearly differentiate between the most significant findings and the auxiliary or side results. This makes it difficult for readers to discern the key contributions and their relevance to the broader field. While the topic itself is intriguing and has substantial potential, the presentation of the material is not sufficiently streamlined, which impacts the readability and overall coherence of the paper. As a result readers may struggle to grasp the paper's main narrative.

2. The paper spends a lot of time comparing $\epsilon$-DP and $(\epsilon, \delta)$-DP, but it doesn’t explain why these two models are chosen or why the differences matter. Some motivation for focusing on these would make the discussion clearer.

3. In Section 3, the Definition 3.3 could have been given more rigorously (especially since it is a definition). I could not understand the second bullet in the first reading, for examples. Also, I don't believe the first-bullet is discussed enough. What does it mean for $v$ to be decided only by the receiver's action? The state of the nature plays on role? In that case it seems that there is no persuasion! Finally, note that $t$ has been already used for the number of receivers, so please change one of two.

Some minor comments:

* Line 261: I would suggest to re-write equation to stay in the margins of the column.

* Line 374, there is an extra fullstop.

* Line 523: The sentence looks odd.

* Line 751: k and m are not in math environment.

* Line 762: "ex-nate" -> ex-ante

* Line 765: missing space

**Questions:**

1. In Section 2.3, Program (2) appears to reduce the exponential number of constraints in Program (1) by leveraging the posterior space and ex ante constraints. Could you expand the discussion on how this alternative formulation is derived from the signal-based formulation? I understand that the objective and the first constraint derive from [31], but the second line of constraints might need some clarification.

2. In Section 3.2, you propose a family of utility functions. Do you believe your results extend to more general utility functions?

**Reviewer Confidence:**

2: The reviewer is willing to defend the evaluation, but it is likely that the reviewer did not understand parts of the paper

**Scope:**

3: The work is somewhat relevant to the Web and to the track, and is of narrow interest to a sub-community

---

### Official Review · Reviewer_KWBa · 2024-12-03

**Novelty:** 5
**Technical Quality:** 7

**Review:**

*Summary*

Bayesian persuasion is a classic framework for studying persuasion in advertising, where a platform shows some signal about their consumer data to convince an advertiser to advertise on the platform, and must decide on a signaling scheme that will yield the best results for the platform. However, it may be important for the platform to preserve customer privacy when revealing this signal. This paper studies the problem of Bayesian persuasion under several different differential privacy constraints. The paper shows that differential privacy and platform utility are fundamentally at odds, characterizes the optimal signaling scheme under DP in terms of concave hulls, and finds a polynomial-time algorithm for computing the optimal signaling scheme.

*Strengths*

- This paper is clearly written, and synthesizes definitions and results from two relevant fields.
- The result that there is a constant utility gap between the sender's maximum utility under $(\epsilon, \delta)$-DP and $\epsilon$-DP for any $\delta$ is very interesting, and implies that (a) ensuring privacy is in some sense "against the platform's financial interests" and (b) that any $\delta > 0$ can be significantly better than $\delta = 0$.
- The mathematical results, in particular the characterization results in Section 4, seem very rich and general.

*Weaknesses*

- The characterizations in Section 4 are mathematically interesting and elegant, and provide conditions (in the corollaries and Appendix B) on when the sender benefits from persuasion. However, it is not clear what are the practical implications of these conditions are. It is also difficult to tell what the implications of Figure 3 are.
- Theorem 5.2, which is the main result showing a polynomial-time algorithm for computing the optimal signaling scheme, holds when Assumption 5.1 is satisfied, but it is difficult to assess the reasonableness of Assumption 5.1.
- A large portion of the results in Section 3 show that "there exists a prior $\mu$" such that the result holds. Exploring the robustness of these results to different choices of $\mu$, for example through further theoretical analysis or experiments, would help readers to understand the implications of these results in real settings.

**Questions:**

(see above for context)
- Do the authors have a sense of how sensitive the worst-case results in Section 3 are to the choice of prior?
- What are the insights on DP in Bayesian persuasion that a reader should take away from Section 4?
- How reasonable is Assumption 5.1?

**Reviewer Confidence:**

2: The reviewer is willing to defend the evaluation, but it is likely that the reviewer did not understand parts of the paper

**Scope:**

3: The work is somewhat relevant to the Web and to the track, and is of narrow interest to a sub-community